# A 24-h activity profile and adiposity among children and adolescents: Does the difference between school and weekend days matter?

**David Janda**[1], **Aleš Gába**[1]*, **Ondřej Vencálek**[2], **Stuart J. Fairclough**[3], **Jan Dygrýn**[1], **Lukáš Jakubec**[1], **Lukáš Rubín**[1]

**1** Faculty of Physical Culture, Palacký University Olomouc, Olomouc, Czech Republic, **2** Faculty of Science, Palacký University Olomouc, Olomouc, Czech Republic, **3** Health Research Institute and Movement Behaviours, Health, and Wellbeing Research Group, Department Sport and Physical Activity, Edge Hill University, Ormskirk, United Kingdom

* ales.gaba@upol.cz

## Abstract

### Background

Twenty-four-hour movement behaviours are gaining attention in the research community. However, no study has addressed how 24-h activity profiles vary between structured and less structured days and whether an unfavourable activity profile is associated with child-hood obesity. We aimed to analyse differences between school day and weekend day 24-h activity profiles and their associations with adiposity indicators among children and adolescents.

### Methods

Participants were 382 children and 338 adolescents who wore wrist accelerometers for 24 hours a day for seven consecutive days. The 24-h activity profile expressed by the average acceleration (AvAcc) and intensity gradient (IG) were estimated from multi-day raw acceler-ometer data. Adiposity indicators included body mass index (BMI) *z*-score, fat mass per-centage (FM%), fat mass index (FMI), and visceral adipose tissue (VAT). Multiple linear regression of activity profile metrics and adiposity indicators was performed separately for school and weekend days.

### Results

Weekend days AvAcc and IG were lower compared to school days in both age groups ($p$ <0.001 for all). Specifically, AvAcc was lower by 9.4% and 11.3% in children and adoles-cents, respectively. IG on weekend days was lower (more negative) by 3.4% in children and 3.1% in adolescents. Among children, on school days AvAcc and IG were negatively associ-ated with FM%, FMI, and VAT, whilst on weekend days AvAcc was positively associated with BMI *z*-score, FMI, and VAT ($p < 0.05$ for all). Among adolescents, negative associations were found between weekend day AvAcc and IG and FM% and FMI ($p < 0.05$ for all), respectively.

**Data Availability Statement:** All data files are available from the Figshare database: https://doi.org/10.6084/m9.figshare.21293385.v1.

**Funding:** AG, JD, LR, and LJ were supported by grant from the Czech Science Foundation (18-09188S). AG, DJ, JD, and OV were supported by grant from the Czech Science Foundation (22-02392S). The funders had no role in study design, data collection and analysis, decision to publish, or preparation of the manuscript.

**Competing interests:** The authors have declared that no competing interests exist.

## Conclusions

This study confirms the importance of 24-h activity profile as a potentially protective factor against excess adiposity. The variability of movement behaviours during structured and less structured days should be considered when optimizing the 24-h movement behaviours to prevent childhood obesity.

## Introduction

The relationship between 24-h movement behaviours and health outcomes is attracting growing attention in public health research [1]. From a movement perspective, the 24-h day is distributed among sleep and waking movement behaviours (i.e., sedentary behaviour [SB] and physical activity [PA]) [2]. A body of evidence suggests that sleep deprivation, extensive SB, and an insufficient level of moderate to vigorous PA are contributors to childhood obesity [3–6]. Nevertheless, most of the previously published studies did not consider the interrelations of these behaviours and studied the health effects of one behaviour in isolation from the others [7, 8]. Thus, recent research has focused on investigating the combined effects to uncover potential interactions between 24-h movement behaviours in the prevention of obesity [4, 9, 10].

The traditional way to study the combined effects of 24-h movement behaviours on obesity is to analyse the daily composition of the amount of time spent in sleep, SB, and PA of different intensities. Such an approach helps to explain only part of the picture as the 24-h movement behaviours construct is multidimensional in its nature [11]. To capture all dimensions of movement behaviours over a day, it is recommended to determine the activity profiles [12]. This should be done using high-resolution accelerometer-based 24-h data that allows researchers to derive continuous data-driven analytical metrics and estimate activity profiles [13]. An example of these metrics is average acceleration (AvAcc), which describes the total volume of PA over the day, and intensity gradient (IG), which captures the entire intensity distribution of activity and provides information about the intensity of activities during the day [13, 14].

The role of the 24-h activity profile in the development of childhood obesity is largely unexplored as there are only a few studies investigating the associations of AvAcc and IG metrics with obesity indicators [14–17]. Besides, it seems appropriate to consider the variation in the activity profile between days of the week, which might have a moderating effect on childhood obesity. The recent meta-analysis by Zosel and colleagues [18] showed that adolescents' movement behaviours are healthier during more structured days. Structured days are typically school days that are characterised by consistent, structured, less autonomous, and segmented days with adult supervision compared to weekend days, which are less structured and take place in a more autonomous environment, providing children with a greater choice in how to spend their time. This aspect of human behaviour is described by the *Structured days hypothesis* (SDH), which was developed to help understand differences in obesogenic behaviour [19].

To date, there is no evidence to test the SDH using continuous data-driven metrics that describe the 24-hour activity profile. Furthermore, it remains unclear whether stability in activity profiles between school and weekend days might be associated with obesity indicators. To help bridge this evidence gap, the present study aimed to verify the SDH by analysing differences in 24-h activity profiles between school and weekend days among children and adolescents. Additional objectives were to explore associations of school and weekend days activity profiles with obesity indicators and to investigate whether the school-weekend activity profile change affects obesity risk.

## Methods

### Study sample

Participants were recruited from seven elementary and four secondary schools across the Czech Republic in 2018 and 2019. A total of 862 children (8–13 years) and adolescents (14–18 years) participated in the current study. Of all participants, 69 had to be excluded because they did not meet the accelerometer wear time criteria, and the other 73 did not undergo the adiposity assessment (S1 Fig). Thus, 720 participants (56% girls) had complete data for all variables of interest and were included in the analytic sample. A detailed description of the recruitment procedure and data collection is available elsewhere [20, 21].

Written informed consent was obtained from the parents or legal guardians. The research was carried out in accordance with the Helsinki ethical declaration for medical research involving human subjects [22]. Ethical approval was granted by the Ethical Committee of the Faculty of Physical Culture, Palacký University Olomouc under register number 19/2017.

### 24-h activity profile

Raw accelerometer data from ActiGraph GT9X Link or wGT3X-BT (ActiGraph, Pensacola, FL, USA) were used to assess the 24-h activity profile. Participants wore an accelerometer on their non-dominant wrist for 24 hours over 7 consecutive days. An accelerometer worn at a non-dominant wrist was proven to be a valid measuring method with high compliance [23, 24]. Monitoring was carried out during weeks with a regular school schedule. Each participant received a monitoring device and was instructed on how to wear it. The devices were collected after eight days after their distribution. The ActiLife software version 6.13.4 (ActiGraph, Pensacola, FL, USA) was used to initialise the accelerometers at 100 Hz sampling frequency on three axes. Raw accelerometer data (gt3x format) were downloaded and converted into.csv files for offline processing using the open-source R package GGIR version 2.1-0 [25]. Default GGIR settings for autocalibration, non-wear time detection and imputation of missing data with average values from the same time points on other days were selected. The average magnitude of dynamic acceleration using Euclidean Norm minus 1 $g$ with negative values rounded up to zero was calculated over a 5-s epoch and expressed in milli gravitational units (m$g$) [26].

The 24-h activity profile metrics were calculated separately for school days and weekend days. The AvAcc reflects a volume of total PA. The IG metric was calculated in GGIR as described by Rowlands et al. [14]. The IG reflects the reduction of time accumulated in PA of higher intensities and is always negative. It represents the slope of the curve of the relationship between $log$ values for intensity and time. School-weekend activity profile change represents the change of values between school and weekend days and was calculated as an absolute value of a relative difference between school and weekend days for AvAcc and IG. Participants were excluded if accelerometer files demonstrated post-calibration error >10 m$g$, they did not meet the wear time criteria, which was set to 3 school days and 1 weekend day with a minimum of 16 hours per day representing a valid day's accelerometer wear [27], or if wear data for each 15-min period in the 24-h cycle were not available.

### Adiposity indicators

Adiposity was assessed by the bioelectrical impedance InBody 720 analyser (Biospace, Seoul, South Korea) using the manufacturers' equations. The multi-frequency bioelectrical impedance analysis is a valid method for assessing adiposity in the target population [28]. Fat mass percentage (FM%) and fat mass index (FMI) were used as an indicator of total adiposity, while visceral adipose tissue (VAT) indicates central adiposity. FMI was calculated by dividing the

amount of fat mass (kg) by body height squared. The participants' adiposity was assessed on the school premises in the morning, the day when the accelerometers were distributed. Participants were asked to come fasting at least 4 hours before the examination and to avoid vigorous PA at least one day before the measurement to ensure a standard measurement procedure. Additionally, participants' body mass index (BMI) was calculated from measured height and weight and was used to estimate sex- and age-specific BMI $z$-score [29]. The measurement was conducted in the school area during standard school hours.

## Confounding factors

Based on our previous research on 24-h movement behaviours [30, 31], the covariates include sex, maternal weight status, maternal education level, and accelerometer wear time. Maternal weight status was calculated based on self-reported BMI. Maternal education level was collected in eight categories and dichotomised before analysis (0 –non-university level, 1 –university level). Missing values for maternal education ($n = 25$) and height ($n = 49$) were imputed by the multiple imputation method with 10 iterations to create five imputed datasets based on the following predictors: maternal and parental education level (dummy variable), school location (dummy variable), BMI, BMI $z$-score, FM%, FMI, and free fat mass index.

## Statistical analysis

The analysis was carried out separately for children and adolescents, as there was an interaction between age groups and activity profile metrics ($p < 0.05$). Before the analysis, potential extreme values were identified and optimised using the winsorizing technique [32]. A descriptive statistic is presented as mean and standard deviation (SD) for continuous variables and as percentages for categorical variables. The differences between school days and weekend 24-h activity profiles were tested using Welch's $t$-test.

Associations between 24-h activity profile metrics (explanatory variable) and adiposity indicators (response variable) were analysed using multivariable linear regression. Regression analysis was performed separately for school (Model 1) and weekend days (Model 2). The school-weekend activity profile change was analysed in Model 3. All regression models were adjusted for confounding factors. Moreover, Models 1 and 2 were adjusted for the activity profile metric during the remaining part of the week (i.e., school days for weekend days and vice-versa). Model 3 was additionally adjusted for the activity profile metric of the entire week. This selection was done to ensure that all the models consisted of the data for the entire week. The variance inflation factor (VIF < 2.6 units for all models) indicated no multicollinearity issue.

Since the interaction between school and weekend days for the AvAcc metric was confirmed in children, the interaction effect was included in Models 1 and 2, and the AvAcc variable was centred to limit collinearity and ensure interpretability of the results. Generalised linear modelling with gamma family was used for models with FMI (response variable) to ensure assumptions for the analysis because a significantly right-skewed distribution of residuals was detected in both age groups.

All analyses were conducted using R software version 4.1.2 (R Foundation for Statistical Computing, Vienna, Austria), and the level of significance was set to $p < 0.05$.

## Results

The descriptive characteristics of the study sample are presented in Table 1. Adolescents had a higher FMI by 0.9 kg/m$^2$ ($p < 0.001$) and VAT by 12.1 cm$^2$ ($p < 0.001$) in comparison with children. The AvAcc for the entire week was lower by 9.0 m$g$ ($p < 0.001$) in adolescents compared to children. A notable difference in intensity distribution was found between age

**Table 1. Descriptive characteristics of children and adolescents.**

| | | Children (n = 382) | | Adolescents (n = 338) | |
|---|---|---|---|---|---|
| | | **Mean** | **SD** | **Mean** | **SD** |
| Age, years | | 11.7 | 1.6 | 16.4 | 1.3 |
| Height, cm | | 151.6 | 12.1 | 170.5 | 8.7 |
| Weight, kg | | 43.8 | 11.5 | 63.7 | 12.0 |
| Adiposity indicators | | | | | |
| | Body mass index z-score | 0.27 | 1.15 | 0.23 | 1.02 |
| | Fat mass percentage, % | 19.5 | 8.2 | 20.7 | 9.2 |
| | Fat mass index, kg/m² | 3.8 | 2.0 | 4.7 | 2.7 |
| | Visceral adipose tissue, cm² | 43.5 | 26.8 | 55.6 | 34.2 |
| 24-h activity profile (all days) | | | | | |
| | Average acceleration, m*g* | 41.6 | 11.2 | 32.6 | 8.2 |
| | Intensity gradient | −2.09 | 0.16 | −2.31 | 0.18 |

SD–standard deviation.

categories as adolescents had a lower (more negative) IG by 0.22 units ($p < 0.001$) than children. Approximately 25% of the children and 22% of the adolescents were classified as overweight or obese. The study sample included 37% of participants whose mothers were classified as overweight or obese and 39% of mothers reported a university degree. No significant differences were observed between age groups in maternal characteristics. The median number of days with valid accelerometer data was 5 and 2 for school days and weekend days, respectively, with an average wear time of 23.6 hours per day.

Table 2 shows the analysis of the differences in the 24-hour activity profiles between school and weekend days. The average values of the activity profile metrics differed significantly between school and weekend days in both age groups ($p < 0.001$ for all). Compared to school days, the AvAcc on weekend days was lower by 9.4% and 11.3% in children and adolescents, respectively. The mean value of the IG for weekend days was lower (more negative) by 3.4% in children and by 3.1% in adolescents compared to school days.

The associations between activity profile metrics and adiposity indicators among children can be found in Table 3. Higher school day AvAcc was associated with a lower FM% ($B = −0.09$; 95% confidence interval [CI]: −0.17, −0.001), FMI ($B = −0.03$; 95% CI: −0.05, −0.01), and VAT ($B = −0.53$; 95% CI: −0.83, −0.23). Higher IG (less negative) on school days was associated

**Table 2. Differences between school and weekend days 24-h activity profile stratified by age category.**

| | | School days | | Weekend days | | Percentage difference | p-value[a] |
|---|---|---|---|---|---|---|---|
| | | **Mean** | **SD** | **Mean** | **SD** | | |
| Children (n = 382) | | | | | | | |
| | Average acceleration, m*g* | 42.7 | 11.6 | 38.7 | 14.7 | −9.4 | **<0.001** |
| | Intensity gradient | −2.07 | 0.16 | −2.14 | 0.21 | −3.4 | **<0.001** |
| Adolescents (n = 338) | | | | | | | |
| | Average acceleration, m*g* | 33.6 | 8.7 | 29.8 | 10.6 | −11.3 | **<0.001** |
| | Intensity gradient | −2.29 | 0.19 | −2.36 | 0.23 | −3.1 | **<0.001** |

SD–standard deviation.

[a]Differences between school and weekend days were tested using Welch's *t*-test.

Boldface values denote significant differences at $p < 0.05$.

**Table 3. Association between 24-h activity profile metrics and adiposity indicators in children (*n* = 382).**

| | | BMI *z*-score | | | Fat mass percentage (%) | | | Fat mass index (kg/m$^2$)$^e$ | | | Visceral adipose tissue (cm$^2$) | | |
|---|---|---|---|---|---|---|---|---|---|---|---|---|---|
| | | *B* | 95% CI | *p-value* | *B* | 95% CI | *p-value* | *B* | 95% CI | *p-value* | *B* | 95% CI | *p-value* |
| School days[a] | | | | | | | | | | | | | |
| | AvAcc[b] | 0.00 | –0.02, 0.01 | 0.653 | **–0.09** | **–0.17, –0.001** | **0.047** | **–0.03** | **–0.05, –0.01** | **0.009** | **–0.53** | **–0.83, –0.23** | **<0.001** |
| | IG | –0.27 | –1.16, 0.61 | 0.544 | –3.71 | –9.81, 2.40 | 0.234 | **–1.79** | **–3.57, –0.02** | **0.033** | **–29.02** | **–50.33, –7.73** | **0.008** |
| Weekend days[c] | | | | | | | | | | | | | |
| | AvAcc[b] | **0.01** | **0.002, 0.02** | **0.019** | 0.08 | –0.01, 0.14 | 0.067 | **0.02** | **0.01, 0.04** | **0.014** | **0.29** | **0.04, 0.54** | **0.025** |
| | IG | 0.02 | –0.67, 0.71 | 0.946 | –1.30 | –6.06, 3.47 | 0.593 | 0.18 | –1.09, 1.42 | 0.775 | –4.19 | –20.79, 12.46 | 0.622 |
| School-weekend activity profile change[d] | | | | | | | | | | | | | |
| | AvAcc | 0.51 | –0.15, 1.16 | 0.130 | 2.02 | –2.51, 6.54 | 0.381 | 0.64 | –0.49, 1.85 | 0.301 | 9.45 | –6.34, 25.23 | 0.240 |
| | IG | 0.17 | –1.82, 2.16 | 0.865 | –2.82 | –16.58, 10.93 | 0.687 | –1.23 | –4.71, 2.61 | 0.497 | –7.22 | –55.26, 40.81 | 0.767 |

AvAcc–average acceleration, *B*–Unstandardized regression coefficient, BMI–body mass index, CI–confidence interval, IG–intensity gradient.

[a] Model was adjusted for sex, maternal weight status, maternal education level, accelerometer wear time and weekend days activity profile metrics.

[b] Model was additionally adjusted for interaction between school and weekend days.

[c] Model was adjusted for sex, maternal weight status, maternal education level, accelerometer wear time and school days activity profile metrics.

[d] Model was adjusted for sex, maternal weight status, maternal education level, accelerometer wear time and activity profile metrics of the entire week.

[e] Generalized linear modelling with gamma family was used.

Boldface values denote significant association at *p* < 0.05.

with lower adiposity represented by FMI (*B* = –1.79; 95% CI: –3.57, –0.02) and VAT (*B* = –29.02; 95% CI: –50.33, –7.73). Significant positive associations were found between weekend day AvAcc and BMI *z*-score score (*B* = 0.01; 95% CI: 0.002, 0.02), FMI (*B* = 0.02; 95% CI: 0.01, 0.04), and VAT (*B* = 0.29; 95% CI: 0.04, 0.54). Negative interactions between school and weekend day AvAcc were identified for all adiposity indicators (S1 Appendix). No associations with adiposity indicators were found for weekend day IG and for the school-weekend activity profile change.

Table 4. shows the results of the regression analysis for adolescents. Weekend day AvAcc was negatively associated with FM% (*B* = –0.10; 95% CI: –0.18, –0.01). Higher weekend day IG (less negative) was associated with lower adiposity represented by FM% (*B* = –5.06; 95% CI: –9.02, –1.09) and FMI (*B* = –1.36; 95% CI: –2.68, –0.04). The regression analysis did not reveal any significant associations for any of the school day activity profile metrics or for the school-weekend activity profile change.

## Discussion

In the present study, we found significantly higher values of activity profile metrics on school days compared to weekend days in both age groups. Our results suggest negative associations between school day AvAcc and IG and adiposity indicators in children. Furthermore, children's weekend day AvAcc was positively associated with adiposity. Among adolescents, negative associations between activity profile metrics and adiposity were identified for weekend days. No associations with adiposity indicators were found for the school-weekend activity profile change in both age categories.

In line with previous studies [14, 33], we observed that children had a better 24-h activity profile than adolescents who had a lower volume and intensity distribution on both school and weekend days. The difference might be explained by the distinct patterns of movement behaviours between these age groups. Children tend to be less sedentary and more active than adolescents, as they are more likely to avoid prolonged sedentary bouts, their SB is more

**Table 4. Association between 24-h activity profile metrics and adiposity indicators in adolescents (*n* = 338).**

| | BMI *z*-score | | | Fat mass percentage (%) | | | Fat mass index (kg/m²)[e] | | | Visceral adipose tissue (cm²) | | |
|---|---|---|---|---|---|---|---|---|---|---|---|---|
| | *B* | 95% CI | *p-value* | *B* | 95% CI | *p-value* | *B* | 95% CI | *p-value* | *B* | 95% CI | *p-value* |
| School days[a] | | | | | | | | | | | | |
| AvAcc[b] | 0.01 | −0.01, 0.02 | 0.461 | −0.05 | −0.16, 0.05 | 0.300 | −0.01 | −0.04, 0.02 | 0.501 | −0.30 | −0.76, 0.16 | 0.198 |
| IG | −0.08 | −0.77, 0.61 | 0.830 | −4.40 | −9.29, 0.48 | 0.077 | −1.19 | −2.82, 0.43 | 0.152 | −20.31 | −42.19, 1.57 | 0.069 |
| Weekend days[c] | | | | | | | | | | | | |
| AvAcc[b] | 0.00 | −0.01, 0.01 | 0.829 | **−0.10** | **−0.18, −0.01** | **0.022** | −0.02 | −0.05, 0.003 | 0.055 | −0.29 | −0.67, 0.09 | 0.128 |
| IG | −0.04 | −0.60, 0.52 | 0.878 | **−5.06** | **−9.02, −1.09** | **0.013** | **−1.36** | **−2.68, −0.04** | **0.044** | −16.56 | −34.34, 1.19 | 0.067 |
| School-weekend activity profile change[d] | | | | | | | | | | | | |
| AvAcc | −0.07 | −0.63, 0.48 | 0.795 | −1.02 | −5.02, 2.99 | 0.618 | −0.55 | −1.62, 0.67 | 0.346 | −4.13 | −21.99, 13.72 | 0.649 |
| IG | 0.77 | −0.99, 2.53 | 0.391 | −1.67 | −14.21, 10.86 | 0.793 | 0.37 | −3.21, 4.22 | 0.845 | 16.49 | −39.54, 72.51 | 0.563 |

AvAcc–average acceleration, *B*–Unstandardized regression coefficient, BMI–body mass index, CI–confidence interval, IG–intensity gradient.

[a] Model was adjusted for sex, maternal weight status, maternal education level, accelerometer wear time and weekend days activity profile metrics.

[b] Model was additionally adjusted for interaction between school and weekend days.

[c] Model was adjusted for sex, maternal weight status, maternal education level, accelerometer wear time and school days activity profile metrics.

[d] Model was adjusted for sex, maternal weight status, maternal education level, accelerometer wear time and activity profile metrics of the entire week.

[e] Generalized linear modelling with gamma family was used.

Boldface values denote significant association at $p < 0.05$.

fragmented, and they spend a greater proportion of time in consecutive PA of different intensities [20, 34, 35]. Our recent longitudinal analysis also revealed that prolonged SB and vigorous PA increased, while time spent in light and moderate PA decreased during the transition from childhood to adolescence [36]. It can also be expected that the adolescents' activity profile could be altered by an increasing amount of school-related sedentary time (i.e., more school classes) or shorter sleep duration.

Our findings regarding the differences in the 24-h activity profile metrics between school and weekend days support the SDH. A greater difference between school and weekend days was found for AvAcc compared to IG. One possible explanation might be the fact that the AvAcc is more sensitive to different sleep duration. Generally, children and adolescents sleep more on weekends compared to school days [37, 38], leading to a compensatory decline in waking behaviours (i.e., SB and PA) during weekend days. This assumption could be supported by studies showing less time spent in SB and PA at all intensities on weekend days compared to school days [39, 40]. Since waking behaviours are characterised by a higher intensity (acceleration) than sleep, its decreased level on weekend days inevitably results in lower values of AvAcc. For this reason, the findings based on AvAcc should be interpreted with caution, as they might provide biased results compared to IG. From this point of view, the decrease in AvAcc due to a longer sleep duration could protect against excess adiposity, as the associations between different sleep characteristics were confirmed in previous research [41].

Significant associations were found between both school and weekend day 24-h activity profile metrics and adiposity indicators in children, while only weekend day activity profile metrics were associated with adiposity in adolescents. Associations involving children's school day activity profiles may imply that children's school days are more structured than adolescents', who are more self-controlled and autonomous in their decisions [42, 43]. More structured days can include more pre-planned PA opportunities, which allows children to choose PA over SB and lead to higher AvAcc and IG (less negative). On the other hand, greater autonomy of adolescents can lead to a preference for SB especially recreational screen time, over PA [19]. In light of present findings, increased participation in pre-planned leisure time activities

during school days (e.g., sports or extracurricular PA) seems to be an appropriate intervention strategy for the prevention of excess adiposity in adolescents.

Findings from the current study are in line with the previously published studies showing favourable associations between 24-h activity profile metrics and adiposity indicators in the paediatric population [14, 15, 17]. Moreover, we confirmed that a favourable activity profile was associated with lower values in adiposity [44, 45]. For example, a 1 SD shift in AvAcc was associated with a 1.4 percent point (i.e., 7% predicted change), and 1.1 percent points (i.e., 5% predicted change) lower FM% in children and in adolescents, respectively. Even lower values were observed for VAT in children. Specifically, VAT was lower by 11% and 9% when school and weekend day AvAcc, respectively, increased by 1 SD. Identical findings were observed for IG. A greater IG was associated with a lower FM% by 6% in adolescents and with lower FMI by 8% and 7% in children and adolescents, respectively (unpublished data). These results indicate that optimisation of 24-h movement behaviours might be a promising strategy to prevent excess adiposity in the paediatric population.

An unexpected finding from our study is that positive associations between weekend AvAcc and adiposity were found in children. The possible explanation might be the presence of a negative interaction between school day and weekend day AvAcc (S1 Appendix) for all adiposity indicators in this age group. The strength of the association between AvAcc on weekend days and adiposity depends on the levels of AvAcc on school days (S1 Appendix). Negative (favourable) associations for weekend day AvAcc were identified only in those whose school day AvAcc was above 1 SD (i.e., the most active individuals). We also found that the majority of active children remained active on weekend days (S1 Appendix). This finding is in line with previous research showing that the most active children maintained their sedentary time and PA levels across the week, while less active children tend to be more sedentary and less active at weekends [40]. Thus, our findings may indicate that despite lower weekend AvAcc, children with a higher volume of activity during school days maintain a sufficient volume of activity on weekends that may protect them against excessive adiposity. Future research should investigate this pattern as this was beyond the scope of the present study.

This is the first study that supports SDH using data-driven analytical metrics representing 24-h activity profile in a relatively large sample, including both children and adolescents. Since most of the previously published studies used BMI as a measure of adiposity, the use of various adiposity indicators is another strength of the current study. Several limitations should also be considered when interpreting our findings. First, our study was cross-sectional, which does not allow us to examine the causal relationship between the activity profile and adiposity. Second, since the cause of obesity is multifactorial, there remain various internal and external factors that have not been considered in the final regression models. Last, the AvAcc is calculated from the entire 24-h cycle and includes sleep (a dominant part of the 24-h composition). Therefore, it may be affected by the different duration of sleep (i.e., longer sleep may result in a lower AvAcc), and some participants might use the extra hours on weekends to get more sleep to compensate for a short duration of sleep during school days (i.e., "social jet lag"), whilst others might sleep less (e.g., watching TV or movies, late-night screen time). This issue might affect the results in various ways and should be considered when interpreting AvAcc in future research.

## Conclusion

The results of this study support SDH by showing that children and adolescents had a favourable 24-h activity profile during school days compared to weekend days. Furthermore, the school day activity profile was negatively associated with several adiposity indicators among

children, while a similar relationship was found for weekend days among adolescents. Our findings provide further support for the SDH and go beyond those studies published previously. Future research may focus on studying how optimising 24-h movement behaviours in different parts of the week might contribute to the prevention of childhood obesity.

## Supporting information

**S1 Checklist. STROBE statement—Checklist of items that should be included in reports of** *cross-sectional studies.*
(DOCX)

**S1 Fig. Participant's exclusion flow chart.**
(PDF)

**S1 Appendix. Interaction between school and weekend day AvAcc.**
(PDF)

## Acknowledgments

The authors thank the participants and schools for their participation in the study.

## Author Contributions

**Conceptualization:** David Janda, Aleš Gába, Stuart J. Fairclough.

**Data curation:** David Janda, Aleš Gába, Ondřej Vencálek, Jan Dygrýn, Lukáš Jakubec, Lukáš Rubín.

**Formal analysis:** David Janda, Ondřej Vencálek.

**Funding acquisition:** Aleš Gába.

**Investigation:** David Janda, Aleš Gába, Ondřej Vencálek, Jan Dygrýn, Lukáš Jakubec, Lukáš Rubín.

**Methodology:** David Janda, Aleš Gába, Ondřej Vencálek, Stuart J. Fairclough, Jan Dygrýn.

**Project administration:** Aleš Gába.

**Resources:** Aleš Gába.

**Supervision:** Aleš Gába, Ondřej Vencálek, Stuart J. Fairclough.

**Visualization:** David Janda.

**Writing – original draft:** David Janda, Aleš Gába.

**Writing – review & editing:** Ondřej Vencálek, Stuart J. Fairclough, Jan Dygrýn, Lukáš Jakubec, Lukáš Rubín.

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
