## [Decision Letter · Decision Letter 0]

13 Mar 2023

PONE-D-22-28651

A 24-h activity profile and adiposity among children and adolescents: Does the difference between school and weekend days matter?

PLOS ONE

Dear Dr. Gába,

Thank you for submitting your manuscript to PLOS ONE. After careful consideration, we feel that it has merit but does not fully meet PLOS ONE’s publication criteria as it currently stands. Therefore, we invite you to submit a revised version of the manuscript that addresses the points raised during the review process.

Some revisions have been suggested along with a follow-up review of your manuscript. Therefore, I invite you to respond to these comments and revise your manuscript.  Please note that choosing to revise and resubmit your manuscript does not guarantee that your manuscript will be accepted.

We look forward to receiving your revised manuscript.

Kind regards,

Adria Muntaner Mas

Academic Editor

PLOS ONE

Reviewers' comments:

Reviewer's Responses to Questions

**Comments to the Author**

1. Is the manuscript technically sound, and do the data support the conclusions?

Reviewer #1: Yes

Reviewer #2: Yes

2. Has the statistical analysis been performed appropriately and rigorously? 

Reviewer #1: Yes

Reviewer #2: Yes

3. Have the authors made all data underlying the findings in their manuscript fully available?

Reviewer #1: Yes

Reviewer #2: Yes

4. Is the manuscript presented in an intelligible fashion and written in standard English?

Reviewer #1: Yes

Reviewer #2: Yes

5. Review Comments to the Author

Reviewer #1: Thank you very much for allowing me to review the article "A 24-h activity profile and adiposity among children and adolescents: Does the difference between school and weekend days matter? ".

This paper is well-written and on an important subject. The manuscript provided rich information about the rationale of this study. A research gap was identified, and the purpose of the study was clearly outlined. Also, most of the data in this manuscript are convincing and presented through well-designed experiments.

A few comments for consideration by the authors:

1. Study sample

In the methods section, kindly enumerate the inclusion and exclusion criteria for choosing your participants. （Creating a flow chart）

2. 24-h activity profile

・This study used two accelerometers. What is the reason for this, and how many are used? （respectively）

・Most studies are attached to the hip and thigh, but why did you put the accelerometers on the wrist?

Also, please describe the measurement method in detail.

3. Adiposity indicators

(Biospace, Soul, South Korea)→(Biospace, Seoul, South Korea)

Reviewer #2: In the manuscript it was analyzed the Structured days hypothesis (specifically, how school days versus weekend days affect adiposity) for children and adolescents using continuous data-driven metrics that describe the 24-hour activity profile. Furthermore, it was analyzed whether stability in activity profiles between school and weekend days is associated with obesity indicators. The paper discuss results from a study at schools in the Czech Republic; specifically, the association of obesity indicators with stability in activity profiles was not confirmed there. The analysis is performed very thoroughly by considering all necessary aspects of data processing, the results of regression analysis were discussed with caution. I have only two minor comments to improve readability of the paper:

l.123: "...were imputed by the multiple imputation method with 10 iterations to create five imputed datasets..." Why 10 iterations (and not e.g. 12)? Why five imputed datasets? Can you provided a reference for the imputation method since both dichotomous and continous variables are imputed together?

l.199: Maybe it would be still beneficial for the readers to explain the abbreviations LPA and MPA.

6. PLOS authors have the option to publish the peer review history of their article (what does this mean?). If published, this will include your full peer review and any attached files.

Reviewer #1: No

Reviewer #2: No

---

## [Author Response · Author response to Decision Letter 0]

6 Apr 2023

Please find attached the revised manuscript, along with a point-by-point response to the reviewers' comments. Our team has carefully addressed each comment and made necessary changes to improve the manuscript's quality. We greatly appreciate the reviewers' time and efforts in providing constructive feedback that has helped us strengthen our work. We hope that the revised manuscript meets the standards of the journal and adequately addresses the reviewers' concerns. Thank you for considering our submission.

---

## [Editor Report · Decision Letter 1]

18 Apr 2023

PONE-D-22-28651R1A 24-h activity profile and adiposity among children and adolescents: Does the difference between school and weekend days matter?PLOS ONE

Dear Dr. Gába,

Thank you for submitting your manuscript to PLOS ONE. After careful consideration, we feel that it has merit but does not fully meet PLOS ONE’s publication criteria as it currently stands. Therefore, we invite you to submit a revised version of the manuscript that addresses the points raised during the review process.

Some revisions have been suggested along with a follow-up review of your manuscript. Therefore, I invite you to respond to these comments and revise your manuscript.  Please note that choosing to revise and resubmit your manuscript does not guarantee that your manuscript will be accepted.

We look forward to receiving your revised manuscript.

Kind regards,

Adria Muntaner Mas

Academic Editor

PLOS ONE
---

## [Editor Report · Decision Letter 2]

5 May 2023

A 24-h activity profile and adiposity among children and adolescents: Does the difference between school and weekend days matter?

PONE-D-22-28651R2

Dear Dr. Gába,

We’re pleased to inform you that your manuscript has been judged scientifically suitable for publication and will be formally accepted for publication once it meets all outstanding technical requirements.

Kind regards,

Adria Muntaner Mas

Academic Editor

PLOS ONE
---

## [Editor Report · Acceptance letter]

9 May 2023

PONE-D-22-28651R2 

A 24-h activity profile and adiposity among children and adolescents: Does the difference between school and weekend days matter? 

Dear Dr. Gába:

I'm pleased to inform you that your manuscript has been deemed suitable for publication in PLOS ONE. Congratulations! Your manuscript is now with our production department. 

Kind regards, 

on behalf of

Dr. Adria Muntaner Mas 

Academic Editor

PLOS ONE